**Data Availability Statement:** Participants in our study did not agree to the data material being shared, which means that there is a legal and

# Experiences with using a mobile application for learning evidence-based practice in health and social care education: An interpretive descriptive study

**Susanne Grødem Johnson**[1]☉*, **Kristine Berg Titlestad**[2], **Lillebeth Larun**[3], **Donna Ciliska**[1,4], **Nina Rydland Olsen**[1]☉

1 Department of Health and Functioning, Faculty of Health and Social Sciences, Western Norway University of Applied Sciences, Bergen, Norway, 2 Department of Welfare and Participation, Faculty of Health and Social Sciences, Western Norway University of Applied Sciences, Bergen, Norway, 3 Division of Health Services, Norwegian Institute of Public Health, Oslo, Norway, 4 Faculty of Health Sciences, McMaster University, Hamilton, Canada

☉ These authors contributed equally to this work.
* susanne.grodem.johnson@hvl.no

## Abstract

### Background

Health and social care students are expected to apply evidence-based practice (EBP). An innovative mobile application, *EBPsteps*, was developed to support learning EBP.

### Aim

The aim of this study was to explore health and social care students' experiences of learning about EBP using the mobile application *EBPsteps* during their clinical placements.

### Methods

An interpretive description approach guided the exploration of student experiences. Four focus groups were conducted with a convenience sample of students from three undergraduate degree programs: occupational therapy, physical therapy, and social education. The constant comparison method was used to categorize and compare the qualitative data.

### Results

Three integrated themes were generated: "triggers for EBP", "barriers to EBP", and "design matters". Information needs, academic requirements, and encouragement from clinical instructors triggered the students to use *EBPsteps*. Lack of EBP knowledge, lack of academic demand, and lack of emphasis on EBP in clinical placement were barriers to using *EBPsteps*. Design issues mattered, as use of the app was motivated by design features such as the opportunity to practice EBP in one place and taking notes in a digital notebook. The use of the app was hindered by anticipation that the use of phones during clinical

ethical restriction on publicly sharing our data. In line with recommendations given by the Norwegian Centre for Research Data (project no. 50425), data are stored on a secure server at Western Norway University of Applied Sciences (HVL). The Research Integrity Officer at HVL, Anne-Mette Somby (Anne-Mette.Somby@hvl.no), may be contacted if queries regarding verifiability of data.

**Funding:** The authors received no specific funding for this work.

**Competing interests:** The authors have declared that no competing interests exist.

placements would be viewed negatively by others and by specific design features, such as unfamiliar icons.

## Conclusions

The students perceived the *EBPsteps* app as a relevant tool for learning EBP, although they also suggested specific changes to the design of the app. Requirements must be embedded in the curriculum to ensure that the app is used. Our findings bring important information to developing and implementing mobile applications as a teaching method in health and social care educations.

## Introduction

Our society's welfare depends on health care practitioners' abilities to adapt to changes in clinical practice [1]. Transformative learning is one learning theory that can support the need to adapt, as its purpose is to produce informed change agents [2]. Transformative learning entails using previous experiences, utilizing critical reflection and questioning, and a willingness to change taken-for-granted assumptions [3, 4]. As such, transformative learning can promote the ability to search, analyze, assess, and synthesize information for decision making [2]. These are skills that health care practitioners need to make informed clinical choices and to practice evidence based. Evidence-based practice (EBP) is an approach that requires the use of the best available evidence from research and integrate it with clinical expertise, patient values, and specific circumstances to make clinical decisions [5, 6].

Health and social care educational programs have commonly incorporated EBP into their curriculum [7, 8]. However, healthcare students have reported critical barriers to apply EBP, such as lack of support from clinical instructors, lack of time, and difficulties in finding research evidence [9–11], and struggling to understand the relevance of EBP [12]. These barriers must be considered when planning for EBP teaching.

Young et al. [13] found that the best teaching strategies for improving EBP were multi-faceted, clinically integrated, interactive, and included learner' assessments. EBP teaching is most effective when integrated across curricula, as opposed to stand-alone courses [5, 13]. Teaching EBP to students increases their EBP knowledge and skills [13], although there are still questions to be answered regarding how to most effectively teach EBP [14].

### Literature review

The use of technology within EBP teaching has increased and has proven to be an effective strategy among undergraduates [15]. A systematic review found that E-learning combined with face-to-face learning improved students' EBP knowledge and skills [14]. Four studies have utilized mobile applications (apps) for EBP teaching among healthcare students [16–19], and two studies among medical students [20, 21]. Three of these studies showed that students who used a mobile device to access EBP resources improved their EBP abilities [18–20]. Carlsen [16] emphasized that few students used their app, even though it was perceived as helpful by those who did. Students who utilized EBP apps during clinical placement reported barriers to use such as concerns about theft, problems with internet connection, and small screen sizes [18]. Students also emphasized EBP as vague and difficult to perform outside the educational environment [17]. A few mobile apps have been developed that support the EBP process. One example is the PubMed Mobile [22], which connects with the PubMed database and supports

only search and retrieval. Another example is the BestEvidence [23], which connects to the Trip search engine and supports search, retrieval, and critical appraisal. However, these apps do not support all steps of the EBP process.

Mobile devices, such as phones and personal digital assistants, are now commonly used in higher education [24]. Systematic reviews have found that mobile devices provide extendable learning environments and access to a wide range of information and learning resources, and they can motivate adaptive and collaborative learning outside the classroom [25–29]. However, utility and features such as small screen size and connectivity problems may pose difficulties when using mobile devices for learning [26, 30]. Nursing students described that low technology literacy and negative reactions from staff and patients with regard to students' use of the device were barriers to using mobile devices [30, 31]. Another systematic review revealed that busy clinical settings, distractions by social connectivity, and unclear policies regarding the use of mobile devices on the clinical unit could also affect the devices' impact [32].

Two systematic reviews emphasized that the success of implementing mobile devices and apps in learning environments depends on the perceived usefulness of the app [25] and well-planned strategies integrated into curricula facilitated by faculty [33]. In order to determine what makes a mobile app usable, it is necessary to examine if the app is considered to be useful, efficient, effective, satisfying, learnable, and accessible [34].

## Development of a mobile application to support learning EBP

An innovative mobile app, *EBPsteps*, was developed at the Western Norway University of Applied Sciences (HVL) to support the learning of EBP among health and social care students as a supplement to other EBP teaching methods (Fig 1).

*EBPsteps* was launched in 2015. Student representatives and faculties from various professions, including physiotherapy, nursing, occupational therapy, social education, and engineering, were involved in its development. The *EBPsteps* app guides users through all the five EBP steps that involve: 1) identifying information needs and formulating answerable questions; 2) finding the best evidence to answer the questions; 3) critically appraising the evidence; 4) applying the results to clinical practice; and 5) evaluating performance [5, 13]. As such, this five-step model assists students' EBP learning process. The app links to the Norwegian Electronic Health Library [35, 36], which includes learning resources for EBP, and access to guidelines, systematic reviews, scientific journals, and a wide variety of other full-text resources. The

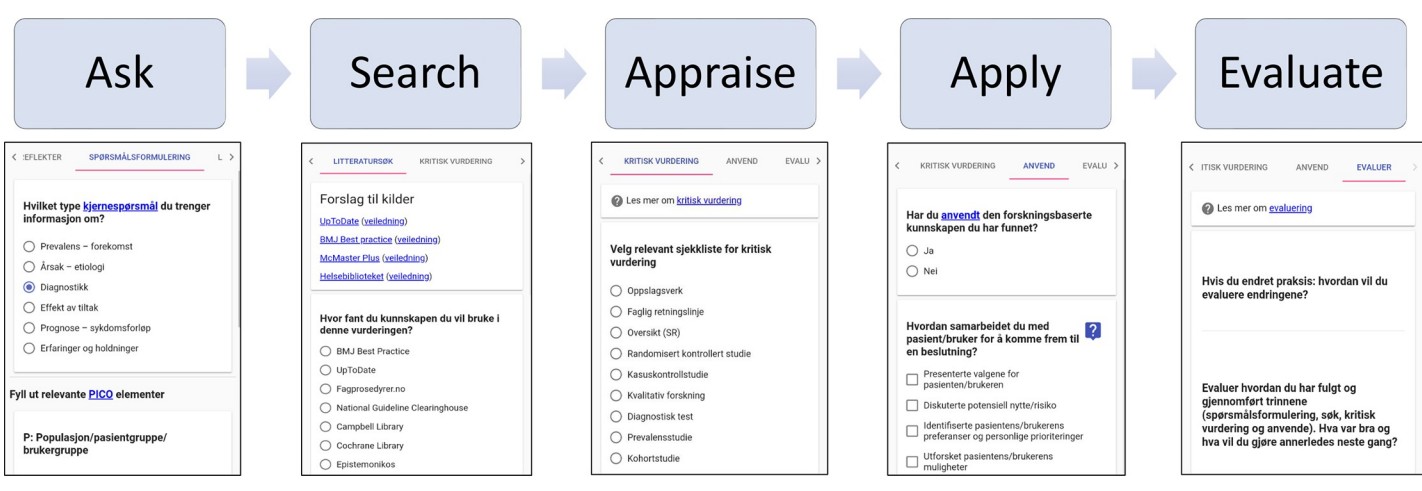

**Fig 1. Sample screen of the *EBPsteps* application.**

*EBPsteps* app allows users to document and save processes related to each EBP step. All information stored in *EBPsteps* can be shared via e-mail, allowing for assessment and feedback from teachers and peers. The app is freely available and can be accessed via any device through the website http://www.ebpsteps.no [37]. While this app is currently only available in Norwegian, we believe that the lessons learned from this study apply to the development of other educational apps. Furthermore, it is planned that *EBPsteps* will be available in other languages with customization of linked resources.

Although technology can facilitate EBP teaching, further research is recommended on how technology can be used to enhance the learning process [15]. The aim of this study was to explore health and social care students' experiences of learning about EBP using the mobile application *EBPsteps* during their clinical placements.

## Materials and methods

### Design and sample

An interpretive description approach guided the process of exploring undergraduate students' experiences related to using the *EBPsteps* app during clinical placements. Interpretive description is regarded as a suitable research strategy to study phenomena in applied disciplines, such as nursing, education, or management [38]. The use of this strategy is relevant when the research aims to generate practical knowledge [39, 40]. The Consolidated criteria for reporting qualitative research (COREQ) checklist were followed to ensure transparent reporting of this study [41].

We introduced the app and its functions to four cohorts of Norwegian undergraduate students: third-year social education (SE) students ($n = 68$) (SE1), third-year occupational therapy (OT) students ($n = 26$), third-year physiotherapy (PT) students ($n = 66$), and second-year SE students ($n = 64$) (SE2) (Table 1).

A member of the research team briefly introduced the app and its functions to the students. Students were encouraged to use the app during their next clinical placements. No further educational support was offered, and there was no assessment or requirements related to the use of *EBPsteps*.

**Table 1. Characteristics of participants.**

| Characteristics | Social Education 1 (3rd year) | Occupational Therapy (3rd year) | Physical Therapy (2nd year) | Social Education 2 (2nd year) |
|---|---|---|---|---|
| Number of students in the cohort | 68 | 26 | 66 | 64 |
| Students invited/participated | 10/4 | 6/3 | 5/2 | 12/6 |
| Gender | | | | |
| Male | 1 | | | |
| Female | 3 | 3 | 2 | 6 |
| Age | | | | |
| 20–29 | 4 | 3 | 1 | 3 |
| 30–39 | | | 1 | 3 |
| Clinical placement | | | | |
| Specialist health service | | 2 | | |
| Primary care/schools | 4 | 1 | 2 | 6 |
| EBP teaching sessions* | 12 | 24 | 37 | 12 |

*Teaching sessions, each lasted 45 minutes

Via e-mail, we purposefully recruited participants among the thirty-three students who chose to use the app during clinical placements: ten SE1 students, six OT students, five PT students, and twelve SE2 students. In total, fifteen students, with a mean age of 26, agreed to participate in focus group interviews: four SE1 students, three OT students, two PT students, and six SE2 students. Most participants were female ($n = 14$), and most had completed clinical placements in primary care settings ($n = 13$). All student cohorts had received EBP training in their respective undergraduate programs. Timetables showed that PT students had received a higher number of EBP training sessions (37 sessions) than the other cohorts. SE students received the lowest number of EBP training sessions (12 sessions). In addition, SE students had received stand-alone sessions, whereas the EBP training was integrated across the three-year curricula in the PT and OT programs.

## Data collection

In line with Krueger and Casey [42], focus groups were used to encourage interaction between participants to explore the different perspectives of using the *EBPsteps* app. Each focus group consisted of students from the same undergraduate degree program. All focus groups took place in meeting rooms at the campus, near the students' teaching environment. Participants received a gift card of 500 NOK (50 EUR) for participating in the study. The interview sessions were conducted between February and May 2017, lasted approximately 1.5 hours, and were digitally recorded.

We used a semi-structured interview guide that reflected the aim of the study and previous research within the field of EBP teaching. Literature inspired the development of the interview guide [5, 13, 42, 43], and it also covered themes on usability attributes, such as ease of use, satisfaction, efficiency, and usefulness [25]. The interview guide is attached as a S1 Appendix. Focus group interviews were led by moderators and co-moderators, who were all experienced EBP teachers. One of the moderators (NRO) initiated the development of the app. The moderators and co-moderators were not involved in teaching or assessment of the students they interviewed. In focus groups, there is a risk that some participants may take a passive role in the discussion or be misunderstood [42]. Therefore, a summary of the main feedback and issues brought up during the interviews was e-mailed to participants for clarification, additions, or comments (member-checking).

## Data analysis

The interpretive description approach guided the analysis process and consisted of the following four phases: transcription, broad coding, comparing and contrasting, and developing themes and patterns [39]. In the first phase, the moderator transcribed the interviews. In the second phase, the three moderators (SGJ, KBT, NRO) read all the transcripts thoroughly, became familiar with the interviews' content, and started broad coding and organizing within each focus group. Microsoft Word [44] was used to support the analysis and manage the results to find similarities and differences across focus groups. To stimulate a coherent interpretation consistent with the interpretive description approach [39], the transcripts' key verbatim segments were collected, rather than coding line-by-line. In the third phase, the same three authors coded and organized the results across the different focus groups related to the research question. Concurrently, the constant comparison analysis [39] commenced, looking for similar and different categories and possible themes within and across interviews. The grouping of the data was as broad as possible to avoid premature closure, as described by Thorne [39]. In the fourth phase, an indication of patterns and themes arose, and these were revised several times until a final decision on the interpretive themes was made. The themes were presented to the other authors for comment and clarification. These four phases enabled

us to gain a comprehensive insight and helped us to consider similarities and differences across the different focus groups. By testing and challenging preliminary interpretations, we aimed to achieve an ordered and coherent result.

### Ethical considerations

The Norwegian Centre for Research Data approved the study (project no. 50425). We obtained written informed consent prior to all interviews. The consent forms, transcripts, and recordings were stored on the research server at the University College to preserve confidentiality. We preserved participants' anonymity by eliminating names and any personal information from transcripts to ensure that participants were not recognizable in the presentation of findings. The transcripts were only available to the three moderators/co-moderators (SGJ, KBT, NRO). The recordings will be deleted upon publication.

## Results

Three integrated themes were generated from the analyses: "triggers for EBP", "barriers to EBP", and "design matters". Students reported that information needs, academic requirements, or encouragement from clinical instructors triggered the use of the *EBPsteps* app. Barriers to use *EBPsteps* included lack of EBP knowledge, lack of academic requirements related to course work, or lack of priority given to EBP during clinical placement. A two-way relationship was found between "triggers for EBP" and "barriers to EBP" (e.g., coursework requirements were both a trigger and a barrier). Students who were required to use EBP were motivated to use the app to learn EBP, whereas students who lacked such coursework requirements reported that this hindered them from using the app.

Similarly, clinical instructors who expected students to apply EBP triggered the use of the app, and students without such expectations did not consider using the app. A one-way inter-relationship was found from "design matters" to "triggers for EBP" and "barriers to EBP". Sub-themes related to "design matters", for example, "all about EBP in one place" and "a good overview of the steps within EBP"; motivated use of *EBPsteps*, and as such, worked as triggers. Design issues, such as unfamiliar icons, or problems using the app during clinical placements, were barriers students faced when using *EBPsteps*. Fig 2 illustrates the relationships between these three integrative themes.

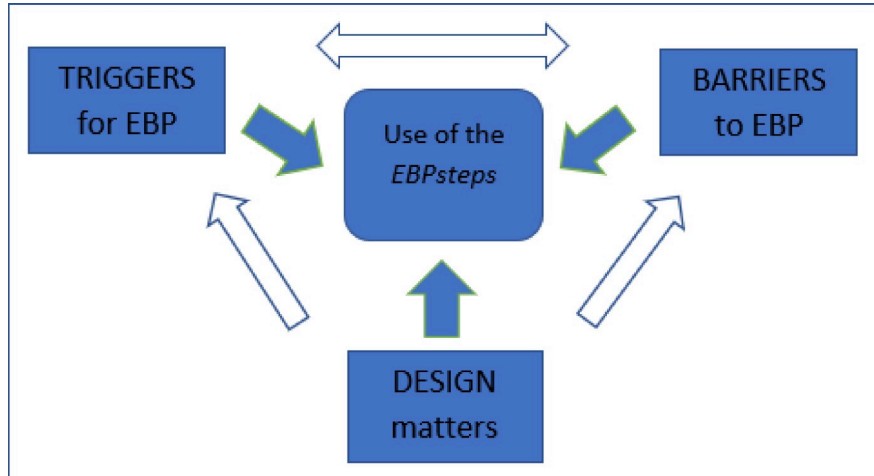

**Fig 2. Patterns and themes related to use of the *EBPsteps*.**

## Triggers for EBP

The use of the *EBPsteps* app was influenced by factors such as the need for information experienced in patient situations, course work requiring the use of research evidence, or clinical instructors expecting students to find and use research evidence.

Students across all interviews reported a need for more knowledge about the various challenges they encountered during clinical placements. Their information needs triggered them to retrieve research evidence and use the app. Several students mentioned that they wanted to find answers to questions about the effectiveness of interventions, such as fall prevention, exercise after pregnancy, or the use of weighted vests for children. As one student said:

> I was on placement in a nursery with children who demonstrated a lot of self-stimulation. The nursery staff used a weighted vest to calm children. Then, I discussed the effect of a weighted vest on self-stimulation with my clinical instructor, and my instructors suggested that I should search for research in that area. Consequently, I used the app (Stud.3, SE1).

The students were motivated to find research evidence, as they needed to justify their choices for the interventions they used in their clinical placements and to inform patients about interventions or treatments. It made a difference to students when they found research evidence. One student said: "Then you've got the answer: okay, it actually means something. You become more confident and start believing it more yourself. This also makes it easier to inform patients, having read that it [the intervention] works" (Stud.2, OT).

Five students from three interviews reported using the app when searching for research for assignments or exams and when they were required to apply EBP skills in clinical placements. Such academic requirements made using the app relevant to students. One student also emphasized that she tried to use the app because she wanted to learn and understand EBP in order to acquire the EBP competence that was required in the program: "I have to, because it [EBP] is a requirement in the take-home exam, and for the undergraduate dissertation and everything, and we must use it. So, I want to learn it. That is why I tried to use the app, because I want to understand it [EBP]" (Stud.3, SE2).

In three of the interviews, students reported situations where clinical instructors encouraged them to apply EBP, which triggered the use of the app. The app was a relevant tool when the students needed to search for research evidence. For example, one student was motivated to search for research evidence via the app when the clinical instructor challenged her to learn more about a specific diagnosis:

> During the clinical placement, I needed to learn more about a diagnosis and a specific challenge related to that diagnosis. I spoke with the clinical instructor, and she suggested that I spend some time investigating this topic. Then I needed to start searching. The app became relevant to use (Stud.1, SE1).

## Barriers to EBP

Across all the interviews, we found several factors that hindered students from using the app during clinical placements, such as lack of EBP knowledge, lack of academic requirements, and instructors not prioritizing EBP.

Some of the students struggled to apply EBP and argued that EBP knowledge was a prerequisite for using the app. They believed their low level of EBP knowledge explained why they

did not use the app, as illustrated by a quote from this student: "It [low level of EBP knowledge] is reflected in our use of the app. When we do not know what is behind it, we cannot do it. We cannot use it [*EBPsteps*], when we do not know anything about it [EBP] (Stud.2, SE2).

Some of the students thought they had to fill in all the elements under each EBP step within the app, which again required more in-depth EBP knowledge. Most students managed to fill in elements of the first two steps, which involved naming a topic and formulating a question, while the remaining steps were mostly not completed. None of the students seemed to understand what to write and why they had to fill in information for the last three EBP steps: the critical appraisal of evidence, application, and evaluation. Some students thought it would be helpful to have explanations of what to do for each step within the app: "I lacked information about how things work. Yes, it was assumed that we knew this from the start" (Stud.3, SE2). Another student commented: "There should have been a demonstration video with examples of how to do it" (Stud.5, SE2).

Two students reported that they had used the app to find information about specific exercises relevant for their clinical placements. As novices, they needed background information about particular therapy sessions and were probably less concerned with research evidence. They felt they had to go through too many steps in the app before they found helpful information for their practice. The students needed precise, practical information about the therapy, and they needed this quickly. As a result, they did not complete all the steps in the app:

> So, I thought it would be wise to look at the latest research and evaluate what is recommended and so forth. That is why I started using the app. However, I have to say that it was a bit too complicated, or not complicated, but there were too many steps before I got useful information because I needed practical, simple exercises [for the patient] (Stud.2, PT).

One student from social education reported no academic requirements or compulsory coursework that required the use of EBP during their clinical placement, and consequently, they did not have any incentive to use the app. As the student said: "I think one of the reasons why I did not write more in the app was that our third year of clinical placement did not require a written assignment" (Stud.1, SE1). Students from the OT and PT programs were required to use EBP in assignments and exams, although without specific documentation that they had followed the EBP steps. One student from physiotherapy worked through the EBP steps of reflection, question formulation, and literature search, which they were asked to do in an assignment. The students were, however, not required to critically appraise or summarize the research results, as illustrated by the following quote: "I only used what I had found to discuss in the assignment. I did not bother to use it [*EBPsteps*] further. I was satisfied because I had what I needed" (Stud.1, PT).

Students across interviews reported few incidences where clinical instructors required them to apply EBP, nor did they state whether their clinical instructors applied EBP. As one student said: "They did not ask about it [EBP] when I was on my clinical placement. The clinical instructors did not talk about it either" (Stud.1, PT).

In three of the interviews, students reported that much of their time during clinical placement was spent on hands-on, patient-related activities such as assessment, treatment, and documentation. EBP was not a priority, and neither was use of the app. After a day at work, one student reported that she was too tired to spend time looking for research. As the student explained: "There was a lot we had to do, document patient treatment after seeing patients, and. . . there was no time to try to find something [research]. There was more we had to do before we finished work. I was very tired when I got home" (Stud.2, PT).

## Design matters

The students reported that some design features of the app motivated or hindered their use of the app. Design aspects that motivated use included the overview of the EBP steps, links to EBP resources, and the digital notebook feature. We also found examples of design-related features that impeded students' use of all the app's functions.

Students across all interviews found the app professional, understandable, transparent, and with a recognizable design. They appreciated that everything they needed to follow the EBP steps could be found in one place. Links to EBP resources were readily available, and students did not have to look for other sources on the web or in books, as reflected in a conversation between two students:

> I also like things to be systematic, and everything we need to work with EBP is in one place. It is very nice (Stud.2, PT). I agree. It is a good idea to have it all in one place so that you do not have to look through x numbers of folders on *itslearning* [digital learning management system]. Where was it again? It was very good. You did not have to browse the book from page to page (Stud.1, PT).

The students liked that they could simultaneously visualize all the EBP steps, which meant that they quickly achieved an overview of the EBP process. Several students found it helpful that links to relevant databases and web pages were provided within the app. Using the checklist integrated with the critical appraisal step in the app helped them understand research design. Students in two of the interviews found the steps understandable, making it straightforward to follow the EBP process. Therefore, the app's design helped students remember the EBP steps and prompted them to use the steps to structure the EBP process. As one student said: "I used it [*EBPsteps*] to make sure that I did it [the EBP process] in the correct order, that I did not skip a step. That is how it [*EBPsteps*] structured the process" (Stud.5, SE2).

The students reported that they used the app to write down ideas and thoughts during their clinical working day, and they appreciated the possibility of exploring the notes later. In this way, the app functioned as a digital notebook. They lacked time to complete all the EBP steps right away, and instead, they wrote down ideas or topics to complete the remaining steps later, as this student explained:

> It was very easy to pick it [*EBPsteps*] up and type in the theme and information needed. Then you could put it [*EBPsteps*] away and continue your search later. Usually, you would not have time to do it right away anyway. So, at least you could write themes down, and you knew where to find them. I found that very useful (Stud.2, SE1).

Although the students were positive about the app and its design, they experienced challenges that hindered use of the app. In one interview, three students explained that new web pages opened when they used links to databases in the app, and they had to make an extra maneuver to return to the app's webpage. The students were not motivated to complete the rest of the EBP steps after they had conducted searches on other web pages: "Yes, it becomes something separate, where the search was in a way detached from the app. For my part, I did not use the app after I had found research [on other web pages], and then it [*EBPsteps*] disappeared in a way" (Stud.3, SE1).

Some students experienced difficulties using the app during their clinical placements. One student explained that using the app felt awkward in the specific setting of their clinical placement:

I did not have the opportunity to use the app in my situation because where I had my clinical placement, we [the student and clinical instructor] supervised the staff in a kindergarten. It might have been possible, but I did not feel that using the app in that situation was the right thing to do. So, for my part, I used the app at home (Stud.3, SE1).

Not all students discovered the app's various functions, such as the vertical three dots icon (. . .) for e-mails, edit button and trash, dictionary, calculator, or the plus sign (+) for creating new appraisals. Students did not realize that their appraisals were stored automatically, nor did everyone understand that they had to scroll down the page to complete all elements at each EBP step. Some students suggested that it would be helpful to have a demonstration video of how to use the app, with examples of how to complete the steps. Several students recommended specific changes to the app's design, as reflected by these students' comments: "I was not aware of the small three dots next to the dictionary" (Stud.2, OT). "Instead of having the three dots, is it possible to have a "letter," a "pen," and a "garbage bin" for example?" (Stud.1, OT).

Some students suggested that we included literature-search tips within the app. They emphasized that searching was difficult and time-consuming, and they did not know where to find databases. Some stated that they thought literature searches seemed easy when demonstrated by the teacher in class, but when they conducted searches themselves, they struggled and found little or nothing of relevance. Therefore, they wanted literature-search tips to be included in the app. Some students also suggested that the app could support their decisions by indicating relevant databases for different questions. One student reported: "Underneath your search, there is only a list [of databases], right. I think, if only they [the questions] could be linked to the databases, [. . .] or somehow be linked together, it would have made it easier" (Stud.3, SE1). Such guidance would have been helpful since the students struggled with identifying relevant databases. They found user manuals for the databases on the university website, but it was difficult for them to choose or find the right manual as there were so many of them.

## Discussion

This study aimed to explore health and social care students' learning experiences about EBP using the mobile application *EBPsteps* during their clinical placements. Mobile applications for higher education must be developed and understood within the relevant educational context. When developing and implementing mobile apps, we need knowledge of how useful, satisfying, learnable, and accessible the users find the app [34]. Themes identified from the data, "triggers for EBP", "barriers to EBP", and "design matters" describe health and social care students' experiences using the *EBPsteps* app during clinical placements. We found that students who used the app were motivated to use it when they perceived it to be useful and relevant for learning EBP. This included if they needed to search for information, they were required to use the app for assignments, or when clinical instructors encouraged them to find or use research evidence. However, factors such as lack of EBP knowledge and lack of requirements to use EBP were perceived as barriers to using the app. The design of the app could both facilitate and hinder its use. The app's design was perceived as helpful as all EBP resources were collected in one place, although some technical issues such as poorly designed icons and navigation issues were perceived as barriers.

Requirements or the lack of requirements for using EBP in assignments worked as triggers or barriers towards EBP and the use of the app. Students reported that they used the app when EBP was required for assignments or when they were encouraged to apply EBP during their clinical placements. By contrast, when students lacked requirements or incentives for using

EBP, they prioritized other competing demands. Several studies report similar findings regarding competing demands and difficulties prioritizing [9, 11, 45, 46]. It seems that when students experience competing demands, they are likely to perceive apps as not useful or relevant and thus choose not to use them. Carlson [16] emphasized that when an app was not necessary for completing an EBP task, the students did not perceive it as relevant. Without incentives or relative advantages, students will most likely not use apps or other interventions [47]. Accordingly, exploring whether users experience apps as relevant and useful is an important part of testing the usability of apps [34].

The extent to which students chose to use the *EBPsteps* app during their clinical placements appeared to depend on whether the clinical instructors expected students to apply research evidence. As such, clinical instructors EBP behavior and their expectations of EBP behavior among students was either a barrier or a trigger towards EBP. In our study, we found that students did not use the *EBPsteps* app, for example, if they did not observe use of EBP by their clinical instructor or if EBP was not expected of them as students. Some clinical instructors encouraged students to apply EBP, and this triggered use of EBP and *EBPsteps*. These clinical instructors were perceived as EBP role models who showed that they understood and valued EBP. The systematic review by Thomas et al. [48] emphasized that clinical instructors should demonstrate, model, and guide students regarding expected EBP skills and behaviors. Ramis et al. [49] also recognized the value of experiencing EBP in clinical practice to motivate students to apply and appreciate the importance of EBP.

In addition to having EBP role models [9, 48, 49], the culture and rules for mobile phone usage at the workplace can influence the utilization of mobile apps [27]. A few of the students were concerned about how using mobile phones in a clinical setting would be perceived by staff and patients. Negative perceptions from health and social care professionals or patients can hinder students from using mobile phones during clinical placements [30], i.e., such perceptions worked as barriers to EBP and the use of the app. Concerns related to theft and the risk of transmitting infection may also restrict phone use [18, 27, 30]. For students to incorporate mobile apps as a learning tool during clinical placements, clinical instructors could advise students of when and how to use apps [27, 32]. The guidance could help balance the use of the app with other activities and support the learning transition. The socio-technical aspect of the mobile app, that is the integration of the app as part of the social world [47], needs to be considered when developing the app. Accessibility of the app is a component of usability testing [34]. The app will be used in a context, and students need support and guidance on when to use the *EBPsteps* during clinical placements.

In our study, a lack of EBP knowledge was reported as a barrier to EBP and use of the app. Previous research into implementing EBP among students recognized that students struggle to apply EBP due to a lack of confidence in their ability to engage with research and lack of time to work in an evidence-based way [11, 12, 45, 46]. To learn how to use EBP, it is recommended that students apply a five-step model [5], and the *EBPsteps* app was developed to support students in this process. Students in our study reported that the app structured the EBP process for them and facilitated the application of research evidence. Other studies have also registered improved EBP abilities among students' who used apps that support EBP [18–20]. Thus, it is likely that using apps that support EBP also helps students become informed change agents who can use their experience and critical thinking skills to question current behaviors in clinical practice [2–4].

Our results showed that design matters, as various design features influenced use of the app. Several of the students interviewed emphasized the possibility of using the app as a digital notebook to track ideas and questions. Queries that are not written down immediately often get lost, so it is useful to have a strategy to rapidly capture and save questions for later retrieval

and searching [6]. One advantage of mobile devices is the possibility to use them when time and place are convenient [26]. Maudsley et al. [32] reported that having a digital notebook is useful for writing down questions and ideas and knowing where to find them. In this respect, *EBPsteps* is a helpful tool where students can gather information about clinical processes and be assisted by valuable e-learning resources integrated into the app. Consequently, apps and other technology are important drivers in transformative learning [2].

Exploring the perceived ease of use of the app is an element of usability testing [25]. Our findings revealed that students in this study struggled to find all the functions in the app, and as such, the design hindered use of the app. Low technology literacy and confidence have been identified as barriers to the use of mobile devices among students [30]. To meet these challenges, introducing the students to the features and interfaces of the application is essential [30, 32]. The successful implementation of technology in education requires technology to be integrated into the curriculum and for teachers to facilitate the use of technology [27, 32, 33]. Further development of the *EBPsteps* app will be necessary to make it even easier for students to use, for example, by automatizing searches for research.

How mobile learning strategies can support learning is worth investigating [28]. The students we interviewed received only a short introduction to the app's functions before their clinical placements, which may explain why some students struggled to identify and use all the functions of the *EBPsteps*. Therefore, future students should receive a more comprehensive introduction. In addition, we believe that active use of the app during various teaching situations, including classroom teaching, could facilitate a better understanding of how and when to use the app. Exposing students to several situations where the app can be used will likely also influence the students' perception of the app's relevance. This approach would be in line with recommendations that state that blended learning is an effective approach to teach EBP [14]. Such actions, where programs mobilize all learning channels, and the use of technology, underline the power of transformative learning [2]. To increase the use of *EBPsteps* during clinical placements, a thorough introduction to the app seems important, including thoughtful planning of how to best implement the *EBPsteps* app into the curriculum and teaching strategies/approaches.

## Methodological considerations

One strength of this study was receiving feedback and experiences from the users to provide insight into how the app may be made more acceptable and helpful to users. Interviewing participants from different undergraduate programs provided different participant categories with various perspectives of experiences using *EBPsteps*. This form of data triangulation [50] allowed us to investigate the data's consistency from multiple perspectives, making the results more trustworthy [51]. By reading and re-reading the interviews, we were able to stay close to participants' contributions and, at the same time, interpret the data. Although we have provided information on how the interpretations were conducted, this is not the only way to interpret the data. Nevertheless, the results are consistent with the participants' descriptions and are an interpretive account of the data. Representative quotes from the data were selected to illustrate the interpretive claims of the data.

When developing a mobile application, it is suggested that the usability of the app should be examined by including five or six users, quickly identifying problems, and then improving the app's design [52]. Our sample consisted of 15 students. Focus groups are regarded as a relevant data collection tool to investigate user experience and are most commonly performed in the early stages of development to evaluate preliminary concepts with representative users [34]. The focus groups took place between two and five months after the students' clinical

placements. This time-lapse may mean that students had forgotten details about using the app. In order to avoid pressuring students to say only positive things about the app, the interview guide included questions regarding both positive and negative experiences (S1 Appendix). As such, we received useful feedback for the further development of the *EBPsteps* app.

## Conclusions

This study aimed to explore health and social care students' experiences of learning about EBP using the mobile application *EBPsteps* during their clinical placements. Three integrated themes described the use of the *EBPsteps*: "triggers for EBP", "barriers to EBP", and "design matters". We found that the students perceived the *EBPsteps* app as a relevant tool for learning EBP, although they also suggested specific changes to the design of the app. The use of the app was triggered by information need in placement, academic requirements, or clinical instructors who required or modeled EBP in placements. When using the app students faced barriers such as lack of EBP knowledge, lack of academic requirements, or EBP not being prioritized during clinical placement. Requirements must be embedded in the curriculum to ensure use of the app. The students' experiences with the *EBPsteps* are also relevant to a broader context of EBP teaching. Our findings bring important information to developing and implementing mobile apps as a teaching method in health and social care educations.

## Supporting information

**S1 Appendix. Focus groups interview guide.**
(DOCX)

## Acknowledgments

The authors wish to thank all the students who participated in this study, making it possible to obtain user experiences of *EBPsteps*.

## Author Contributions

**Conceptualization:** Nina Rydland Olsen.

**Data curation:** Susanne Grødem Johnson, Nina Rydland Olsen.

**Formal analysis:** Susanne Grødem Johnson, Kristine Berg Titlestad, Nina Rydland Olsen.

**Funding acquisition:** Nina Rydland Olsen.

**Investigation:** Susanne Grødem Johnson, Kristine Berg Titlestad, Nina Rydland Olsen.

**Methodology:** Susanne Grødem Johnson, Nina Rydland Olsen.

**Project administration:** Susanne Grødem Johnson, Nina Rydland Olsen.

**Resources:** Susanne Grødem Johnson, Kristine Berg Titlestad, Nina Rydland Olsen.

**Supervision:** Lillebeth Larun, Donna Ciliska, Nina Rydland Olsen.

**Validation:** Nina Rydland Olsen.

**Visualization:** Susanne Grødem Johnson, Nina Rydland Olsen.

**Writing – original draft:** Susanne Grødem Johnson, Nina Rydland Olsen.

**Writing – review & editing:** Susanne Grødem Johnson, Kristine Berg Titlestad, Lillebeth Larun, Donna Ciliska, Nina Rydland Olsen.

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
