## [Decision Letter · Decision Letter 0]

16 Mar 2021

PONE-D-20-40199

Experiences with using a mobile application for learning evidence-based practice in health and social care education: An interpretive descriptive study

PLOS ONE

Dear Dr. Johnson,

Thank you for submitting your manuscript to PLOS ONE. After careful consideration, we feel that it has merit but does not fully meet PLOS ONE’s publication criteria as it currently stands. Therefore, we invite you to submit a revised version of the manuscript that addresses the points raised during the review process.

We look forward to receiving your revised manuscript.

Kind regards,

Gwo-Jen Hwang

Academic Editor

PLOS ONE

Reviewers' comments:

Reviewer's Responses to Questions

**Comments to the Author**

1. Is the manuscript technically sound, and do the data support the conclusions?

Reviewer #1: Yes

Reviewer #2: Yes

2. Has the statistical analysis been performed appropriately and rigorously? 

Reviewer #1: No

Reviewer #2: Yes

3. Have the authors made all data underlying the findings in their manuscript fully available?

Reviewer #1: Yes

Reviewer #2: Yes

4. Is the manuscript presented in an intelligible fashion and written in standard English?

Reviewer #1: Yes

Reviewer #2: Yes

5. Review Comments to the Author

Reviewer #1: This paper aims to investigate the impacts of Experiences with using a mobile application for learning evidence-based practice in health and social care education: An interpretive descriptive study. It's quite interesting. I agree with the authors’ point that this is an EBP health and social care education issue. However, my overall impression of this work is that it is not ready for publication owing to the following reasons:

Figure 1, " Sample screen of the EBPsteps application " it would be more clear and contribution add related EBP step by step, for example, five EBP steps, should figure out E screen, B screen, P screen.

Page 4, line 80~82: “EBPsteps was launched in 2015. Student representatives and faculties from various professions, including physiotherapy, nursing, occupational therapy, social education and engineering, were involved in its development.” it would be better to cite relevant references instead.

Page 4, line 82~83: “The EBPsteps app guides users through the five EBP steps. ” It would be better to add more detail about the five EBP steps and to cite relevant references instead.

Page 4, line 87~89: “The app is freely available and can be accessed via any device through the website http://www.ebpsteps.no . ” For more reader and contrition to improve EBP, why not chose the global popular “PubMed Mobile” app.

Page 5, line 93~94: “Although technology can facilitate EBP teaching, there is still little research in the area of EBP and mobile apps. ” But I search more then the references below that had discussed related issues, so suggest you describe more your background about your research aim and why it's important.

Morris, J., & Maynard, V. (2010). Pilot Study to Test the Use of a Mobile Device in the Clinical Setting to Access Evidence‐Based Practice Resources. Worldviews on Evidence‐Based Nursing, 7(4), 205-213.

Lam, C. K., Schubert, C. F., & Brown, C. A. (2018). Opportunities and Barriers to Building EBP Skills in the Clinical Setting via Mobile Technology.

Carlson, K. W. (2018). Perceptions of an EBP Module Mobile Application by New Graduate Nurses.

Page 5~ 6, table1: Participants “… from three different health and social care programmes: occupational therapy (OT), physiotherapy (PT), and social education (SE). We interviewed participants from different programmes to ensure a diversity

of experiences related to using the EBPsteps app. The app was introduced to four cohorts of students: second and third-year SE students, third-year OT students, and third-year PT students (Table 1).” So discrepancy EBP teaching sessions*, authors how to ensure their EBP education equal and interview qualitative validity.

Page 7, line 93~94: “We used a semi-structured interview guide that reflected the aim of the study and previous.” it would be better to cite relevant interview guide references instead and for contribution, suggestion authors add the study your detail interview guide.

Page 7, line 149~155: “Microsoft Word (23) was used to support the analysis.” It would be better to add detail about the interview analysis. Authors how to get the Microsoft Word results and analysis.

Discussion section, suggest you add your finding linked with transformative

Theory, results sections and discuss with Morris & Maynard (2010) and Carlson (2018) and Lam, Schubert,and Brown (2018).

Most of the references are too old.

Thank you for your efforts.

Reviewer #2: The topic of this research is interesting; however, the paper is not well organized and written.

1. There are several subsections with only one or two paragraphs, which should be avoided.

2. The authors need to reorganize the paper following the standard structure:

Introduction (i.e., background and motivation of the present study)

Literature Review

Development of a mobile application to support learning EBP

Method (or Research design)

Results

Discussion

Conclusion

3. It is suggested that the authors refer to those studies related to the use of mobile technologies in health care or nursing education to strengthen the background and motivation of the present study. Some recommended references are listed as follows:

Chang, C. Y., Kao, C. H., & Hwang, G. J. (2020). Facilitating students’ critical thinking and decision making performances: A flipped classroom for neonatal health care training. Educational Technology & Society, 23(2), 32-46.

Chang, C. Y., Gau, M. L., Tang, K. Y., & Hwang, G. J. (2021). Directions of the 100 most cited nursing student education research: A bibliometric and co-citation network analysis. Nurse Education Today, 96, 104645.

Chang, C. Y., Lai, C. L., & Hwang, G. J. (2018). Trends and research issues of mobile learning studies in nursing education: a review of academic publications from 1971 to 2016. Computers & Education,116, 28-48.

Wu, P. H., Hwang, G. J., Su, L. H., & Huang, Y. M. (2012). A context-aware mobile learning system for supporting cognitive apprenticeships in nursing skills training. Educational Technology & Society, 15(1), 223-236.

6. PLOS authors have the option to publish the peer review history of their article (what does this mean?). If published, this will include your full peer review and any attached files.

Reviewer #1: No

Reviewer #2: No

---

## [Author Response · Author response to Decision Letter 0]

10 May 2021

Response to Reviewers 28th of April 2021

Gwo-Jen Hwang

Academic Editor

PLOS ONE

Dear Dr., Gwo-Jen Hwang

Thank you for the opportunity to submit a revised version of our manuscript. We appreciate the constructive criticism and helpful feedback from all of you. A response to each point is provided below. When editing, we also saw the need to make some other minor changes (e.g., spelling). The article will be uploaded with track changes, making it easy to see its modifications. 

PONE-D-20-40199

Experiences with using a mobile application for learning evidence-based practice in health and social care education: An interpretive descriptive study

PLOS ONE

Response: We have read PLOS ONE's style requirements and updated them to be in line with the journal's requirements.

Response: For privacy reasons, it is impossible to provide access to interview data from this project. We did not ask participants for approval for sharing the data. It is also challenging to anonymize the participants in the transcripts. Therefore, due to legal restrictions on sharing the data, it is impossible to share this data. 

Comments to the Author

1. Is the manuscript technically sound, and do the data support the conclusions?

Reviewer #1: Yes

Reviewer #2: Yes

Response: We appreciate this feedback. ________________________________________

2. Has the statistical analysis been performed appropriately and rigorously? 

Reviewer #1: No

Reviewer #2: Yes

Response: We did not use any statistical analysis in this manuscript since we have used a qualitative method. ________________________________________

3. Have the authors made all data underlying the findings in their manuscript fully available?

Reviewer #1: Yes

Reviewer #2: Yes

Response: We appreciate this feedback.________________________________________

4. Is the manuscript presented in an intelligible fashion and written in standard English?

Reviewer #1: Yes

Reviewer #2: Yes

Response: We appreciate this feedback.________________________________________

5. Review Comments to the Author

Please use the space provided to explain your answers to the questions above. You may also include additional comments for the Author, including concerns about dual publication, research ethics, or publication ethics. (Please upload your review as an attachment if it exceeds 20,000 characters)

 

Response to reviewer 1 

Comments Response

Reviewer #1: This paper aims to investigate the impacts of Experiences with using a mobile application for learning evidence-based practice in health and social care education: An interpretive descriptive study. It's quite interesting. I agree with the authors' point that this is an EBP health and social care education issue. However, my overall impression of this work is that it is not ready for publication owing to the following reasons: 

Response: We appreciate the effort made by both external reviewers and have done our best to respond to each comment where you point to required changes. We refer to page and line in the revised manuscript (version without track changes). 

Figure 1, " Sample screen of the EBPsteps application " it would be more clear and contribution add related EBP step by step, for example, five EBP steps, should figure out E screen, B screen, P screen. 

Response: We appreciate this comment and have corrected it as suggested, page 5, line 104. 

Page 4, line 80~82: "EBPsteps was launched in 2015. Student representatives and faculties from various professions, including physiotherapy, nursing, occupational therapy, social education and engineering, were involved in its development." it would be better to cite relevant references instead. 

Response: We describe people who participated in the development of the EBPsteps app. We do not perceive it as relevant to cite references related to this description, page 5, line 106-108. 

Page 4, line 82~83: "The EBPsteps app guides users through the five EBP steps." It would be better to add more detail about the five EBP steps and to cite relevant references instead. 

Response: We have presented the five EBP steps with references, page 5, line 109-111.

Page 4, line 87~89: "The app is freely available and can be accessed via any device through the website http://www.ebpsteps.no ." For more reader and contrition to improve EBP, why not chose the global popular "PubMed Mobile" app. PubMed Mobile app relates mainly to the search and retrieval of research articles (two of the EBP steps). 

Response: The EBPsteps were developed to support the learning process of EBP, following the work process of all five EBP steps. Therefore, it was necessary to create a new app. A more thorough explanation is now presented in the article, page 5, line 108-112. 

Page 5, line 93~94: "Although technology can facilitate EBP teaching, there is still little research in the area of EBP and mobile apps." But I search more then the references below that had discussed related issues, so suggest you describe more your background about your research aim and why it's important.

Morris, J., & Maynard, V. (2010). Pilot Study to Test the Use of a Mobile Device in the Clinical Setting to Access Evidence‐Based Practice Resources. Worldviews on Evidence‐Based Nursing, 7(4), 205-213.

Lam, C. K., Schubert, C. F., & Brown, C. A. (2018). Opportunities and Barriers to Building EBP Skills in the Clinical Setting via Mobile Technology.

Carlson, K. W. (2018). Perceptions of an EBP Module Mobile Application by New Graduate Nurses. 

Response: We appreciate the suggested studies; they are all relevant. The background has been elaborated, and we have included information from these studies, page 4, line 71-83.

Page 5~ 6, table1: Participants "… from three different health and social care programmes: occupational therapy (OT), physiotherapy (PT), and social education (SE). We interviewed participants from different programmes to ensure a diversity

of experiences related to using the EBPsteps app. 

Response: The app was introduced to four cohorts of students: second and third-year SE students, third-year OT students, and third-year PT students (Table 1)." So discrepancy EBP teaching sessions*, authors how to ensure their EBP education equal and interview qualitative validity. It was relevant to examine the student's experiences of using the app based on different EBP teaching experiences, so a diversity of experiences could be uncovered within the focus groups. This was a strength of this study. The different levels of EBP teaching experience were not something the authors could control, page 6, line 135-138.

Page 7, line 93~94: "We used a semi-structured interview guide that reflected the aim of the study and previous." it would be better to cite relevant interview guide references instead and for contribution, suggestion authors add the study your detail interview guide. We have included literature that inspired the interview guide's development and supplied an explanation of usability attributes included in the interview guide. 

Response: We have clarified that the S1 file is the attachment of the interview guide, page 8, line 165-169. 

Page 7, line 149~155: "Microsoft Word (23) was used to support the analysis." It would be better to add detail about the interview analysis. Authors how to get the Microsoft Word results and analysis. 

Response: We have elaborated how Microsoft Word was used to support the analysis, page 9, line 183. 

Discussion section, suggest you add your finding linked with transformative

Theory, results sections and discuss with Morris & Maynard (2010) and Carlson (2018) and Lam, Schubert,and Brown (2018). 

Response: We appreciate this suggestion. The transformative theory is included both in the introduction, page 3, line 48-57, and in the discussion sections, page 20, line 457-459, and line 466-468. 

The suggested literature is also included in the discussion, page 18, line 420-421 + page 19, line 440-441 + page 20, line 455-456. 

Most of the references are too old.

Response: We have updated the literature review with newer references. We believe the included references are relevant to our article, although we have deleted some of the oldest ones. 

 

Response to reviewer 2 

Comments Response

Reviewer #2: The topic of this research is interesting; however, the paper is not well organized and written. We appreciate the effort made by both external reviewers and have done our best to respond to each comment where you point to required changes. We refer to page and line in the revised manuscript (version without track changes).

1. There are several subsections with only one or two paragraphs, which should be avoided. 

Response: Several subtitles have been removed throughout the manuscript.

2. The authors need to reorganize the paper following the standard structure:

Introduction (i.e., background and motivation of the present study)

Literature Review

Development of a mobile application to support learning EBP

Method (or Research design)

Results

Discussion

Conclusion 

Response: We appreciate this comment and have included a literature review to the introduction, pages 3-4, lines 68-99. We also ensured that the other parts were organized as suggested. 

3. It is suggested that the authors refer to those studies related to the use of mobile technologies in health care or nursing education to strengthen the background and motivation of the present study. Some recommended references are listed as follows:

Chang, C. Y., Kao, C. H., & Hwang, G. J. (2020). Facilitating students' critical thinking and decision making performances: A flipped classroom for neonatal health care training. Educational Technology & Society, 23(2), 32-46.

Chang, C. Y., Gau, M. L., Tang, K. Y., & Hwang, G. J. (2021). Directions of the 100 most cited nursing student education research: A bibliometric and co-citation network analysis. Nurse Education Today, 96, 104645.

Chang, C. Y., Lai, C. L., & Hwang, G. J. (2018). Trends and research issues of mobile learning studies in nursing education: a review of academic publications from 1971 to 2016. Computers & Education,116, 28-48.

Wu, P. H., Hwang, G. J., Su, L. H., & Huang, Y. M. (2012). A context-aware mobile learning system for supporting cognitive apprenticeships in nursing skills training. Educational Technology & Society, 15(1), 223-236. 

Response: We appreciate the suggested studies. We have used systematic reviews (SR) as a higher level of synthesis to inform the background, except with the presentation of mobile apps and EBP teaching. Therefore, we have included the suggested SR from your list of studies, page 4, line 85-87 + page 21, line 478.

---

## [Decision Letter · Decision Letter 1]

2 Jun 2021

PONE-D-20-40199R1

Experiences with using a mobile application for learning evidence-based practice in health and social care education: an interpretive descriptive study

PLOS ONE

Dear Dr. Johnson,

Thank you for submitting your manuscript to PLOS ONE. After careful consideration, we feel that it has merit but does not fully meet PLOS ONE’s publication criteria as it currently stands. Therefore, we invite you to submit a revised version of the manuscript that addresses the points raised during the review process.

We look forward to receiving your revised manuscript.

Kind regards,

Gwo-Jen Hwang

Academic Editor

PLOS ONE

Journal Requirements:

Reviewers' comments:

Reviewer's Responses to Questions

**Comments to the Author**

1. If the authors have adequately addressed your comments raised in a previous round of review and you feel that this manuscript is now acceptable for publication, you may indicate that here to bypass the “Comments to the Author” section, enter your conflict of interest statement in the “Confidential to Editor” section, and submit your "Accept" recommendation.

Reviewer #1: (No Response)

Reviewer #2: All comments have been addressed

2. Is the manuscript technically sound, and do the data support the conclusions?

Reviewer #1: No

Reviewer #2: (No Response)

3. Has the statistical analysis been performed appropriately and rigorously? 

Reviewer #1: No

Reviewer #2: Yes

4. Have the authors made all data underlying the findings in their manuscript fully available?

Reviewer #1: No

Reviewer #2: Yes

5. Is the manuscript presented in an intelligible fashion and written in standard English?

Reviewer #1: Yes

Reviewer #2: Yes

6. Review Comments to the Author

Reviewer #1: Dear Authors,

Thank you for your revised work, but I suggest some advice below,

1.Page 6, line 124-126, “15 students with a mean age of 26 participated in focus group interviews between February and May 2017. Most participants were female 126 (n=14), and most had completed clinical placements in primary care settings (n=13” The Participants section should be rewording based on your Table 1, Table 1. Characteristics of participants not match, for example, Social Education 1, Number of students in the cohort (N=68), but why age only N=4? please check.

in addition, why only 15 students with a mean age of 26 participated in the focus group get an interview? It's vague.

2. Page 9, Results section, should be rewording too because, from table 2, it's difficult to get the related information “Three integrated themes were generated from the analyses: "triggers for EBP", "barriers to EBP", and "design matters".

Besides abstract and results stated that get three integrated themes "triggers for EBP", "barriers to EBP", and "design matters". But exactly, the result section not only three integrated themes, including Triggers for EBP, Information needs in clinical placement, Academic requirements, Barriers to EBP, Lack of EBP knowledge, Lack of academic requirement, Clinical instructors did not require students to apply EBP, EBP was not a priority, Design matters, Motivating design aspects, and Design factors that hindered the use of the app. Suggestion authors amendments, Meanwhile, need to follow your interview guideline.

3. The discussion and conclusions section should follow the Results section too.

Reviewer #2: (No Response)

---

## [Author Response · Author response to Decision Letter 1]

18 Jun 2021

Response to Reviewers 18th of June 2021

Gwo-Jen Hwang

Academic Editor

PLOS ONE

Dear Dr., Gwo-Jen Hwang

Thank you for the opportunity to submit a second revised version of our manuscript. We appreciate the feedback from all of you. A response to each point is provided below. The article will be uploaded with track changes, making it easy to see its modifications. The version with track changes is where we point to the page and line for changes. 

PONE-D-20-40199R1

Experiences with using a mobile application for learning evidence-based practice in health and social care education: an interpretive descriptive study

PLOS ONE

Journal Requirements:

Response: We have reviewed the reference list and made sure that it is complete and correct. None of the cited papers were deleted from the manuscript during this review process. We have not cited papers that have been retracted. 

Reviewers' comments:

Reviewer's Responses to Questions

Comments to the Author

1. If the authors have adequately addressed your comments raised in a previous round of review and you feel that this manuscript is now acceptable for publication, you may indicate that here to bypass the "Comments to the Author" section, enter your conflict of interest statement in the "Confidential to Editor" section, and submit your "Accept" recommendation.

Reviewer #1: (No Response)

Reviewer #2: All comments have been addressed

Response: We appreciate the feedback from reviewer #2 from the previous round of review.

2. Is the manuscript technically sound, and do the data support the conclusions?

Reviewer #1: No

Reviewer #2: (No Response)

Response: As Reviewer #2 has no response regarding this issue, we believe that Reviewer #2 is of the opinion that the manuscript is technically sound and that the data support our conclusion. We notice that Reviewer #1 response is "no", although there is no detailed information about what is missing. We hope that changes we have made to the manuscript regarding other comments (item 6 in the list "6. Review Comments to the Author") are sufficient to ensure that the manuscript is technically sound and that the data support our conclusion. 

3. Has the statistical analysis been performed appropriately and rigorously? 

Reviewer #1: No

Reviewer #2: Yes

Response: As this is a qualitative study, we did not use any statistical analysis in this manuscript.

4. Have the authors made all data underlying the findings in their manuscript fully available?

Reviewer #1: No

Reviewer #2: Yes

Response: Participants in our study did not agree to the data material being shared, which means that there is a legal and ethical restriction on publicly sharing our data. In line with recommendations given by the Norwegian Centre for Research Data (project no. 50425), data are stored on a secure server at Western Norway University of Applied Sciences (HVL). The Research Integrity Officer at HVL, Anne-Mette Somby (Anne-Mette.Somby@hvl.no), may be contacted if queries regarding verifiability of data. ________________________________________

5. Is the manuscript presented in an intelligible fashion and written in standard English?

Reviewer #1: Yes

Reviewer #2: Yes

Response: We appreciate this feedback.

6. Review Comments to the Author

From Reviewer #1:

Dear Authors,

Thank you for your revised work, but I suggest some advice below,

1.Page 6, line 124-126, "15 students with a mean age of 26 participated in focus group interviews between February and May 2017. Most participants were female 126 (n=14), and most had completed clinical placements in primary care settings (n=13" The Participants section should be rewording based on your Table 1, Table 1. Characteristics of participants not match, for example, Social Education 1, Number of students in the cohort (N=68), but why age only N=4? please check.

in addition, why only 15 students with a mean age of 26 participated in the focus group get an interview? It's vague.

Response: We appreciate this advice from Reviewer #1 and have made the following changes to the manuscript: 

- When we first introduce Table 1, we state how many students in the different cohorts were introduced to the app, page 6, lines 136-140 (Table 1, line 1). 

- Thirty-three students used the app during clinical placement. How many students from each cohort using the app is presented on page 7, lines 153-155 (Table 1, line 2).

- In total, fifteen students agreed to participate. We have written this clearer in the text, page 7, lines 156-158 (Table 1, line 2). 

- We reported the student's age on those who participated in this study. 

From Reviewer #1:

2. Page 9, Results section, should be rewording too because, from table 2, it's difficult to get the related information "Three integrated themes were generated from the analyses: "triggers for EBP", "barriers to EBP", and "design matters".

Besides abstract and results stated that get three integrated themes "triggers for EBP", "barriers to EBP", and "design matters". But exactly, the result section not only three integrated themes, including Triggers for EBP, Information needs in clinical placement, Academic requirements, Barriers to EBP, Lack of EBP knowledge, Lack of academic requirement, Clinical instructors did not require students to apply EBP, EBP was not a priority, Design matters, Motivating design aspects, and Design factors that hindered the use of the app. Suggestion authors amendments, Meanwhile, need to follow your interview guideline.

Response: We appreciate this advice from Reviewer #1. Based on this comment, we realized that some clarification was needed. The aim with table 2 was to illustrate the part of the analysis process that involves comparing and contrasting and developing themes. However, this was not clarified sufficiently in the text. We realize that keeping the table could cause confusion and misunderstandings and illustrating all four phases meaningfully in a table is difficult. Therefore, we decided to delete Table 2 from the text. We can however provide this table as an appendix. 

Regarding the comment that "the results section should be rewording". The three themes consist of different explanations for each theme presented in the theme's introduction in the results section. For instance, the theme "triggers for EBP" is explained with "The use of the EBPsteps app was influenced by factors such as the need for information experienced in patient situations, course work requiring the use of research evidence, or clinical instructors expecting students to find and use research evidence." Thus, it was different aspects that could trigger EBP and the use of the EBPsteps app, e.g., information needs in clinical placement, academic requirements, and clinical instructors expect students to apply EBP.

Regarding the comment that we "need to follow your interview guideline". We are unsure how to interpret this comment, and we did not make any changes. The interview guide was developed to support the interviews of the focus groups, and it was not used as a guide for the analysis. We used an inductive process, where the information from the participants guided the analysis process. 

From Reviewer #1:

3. The discussion and conclusions section should follow the Results section too.

Response: We appreciate this advice from Reviewer #1. We have now clarified the link to the themes that emerged: page 19, lines 425-426, and lines 439-440 + page 20, lines 453-454 and line + page 21, line 474 and lines 485-486. 

To ensure that the conclusion aligns with the Results section, we have revised the conclusion with additional explanations related to the Results section, page 23, lines 530-532 and 534-538. 

Reviewer #2: (No Response)

---

## [Editor Report · Decision Letter 2]

24 Jun 2021

Experiences with using a mobile application for learning evidence-based practice in health and social care education: an interpretive descriptive study

PONE-D-20-40199R2

Dear Dr. Johnson,

We’re pleased to inform you that your manuscript has been judged scientifically suitable for publication and will be formally accepted for publication once it meets all outstanding technical requirements.

Kind regards,

Gwo-Jen Hwang

Academic Editor

PLOS ONE
---

## [Editor Report · Acceptance letter]

28 Jun 2021

PONE-D-20-40199R2 

Experiences with using a mobile application for learning evidence-based practice in health and social care education: an interpretive descriptive study 

Dear Dr. Johnson:

I'm pleased to inform you that your manuscript has been deemed suitable for publication in PLOS ONE. Congratulations! Your manuscript is now with our production department. 

Kind regards, 

on behalf of

Dr. Gwo-Jen Hwang 

Academic Editor

PLOS ONE